## Original Research Article

ABA; biophysical model; climate change; fruit development; phytohormone.

**Corresponding author:**
Sun Woo Chung;
Email: sunwchung@jnu.ac.kr

**Associate Editor:**
Dr. George Bassel

# An integrative process-based model of fruit growth as a function of carbon and water fluxes modulated by endogenous abscisic acid in blueberry fruit

Sun Woo Chung[1,2] ![ORCID], Kyungdahm Yun[3] and Soo-Hyung Kim[1]

[1]School of Environmental and Forest Sciences, College of the Environment, University of Washington, Seattle, WA, USA; [2]Department of Horticulture, College of Agricultural and Life Sciences, Chonnam National University, Gwangju, Republic of Korea; [3]Department of Smart Farm, Jeonbuk National University, Jeonju, Republic of Korea

## Abstract

Fruit growth is driven by the interaction of environmental cues and phytohormonal signals. Biophysical models have captured the general trend of fruit growth but often overlook the regulatory role of phytohormones. This study integrates a biophysical framework with the quantitative response of endogenous abscisic acid (ABA) in fruit. ABA dynamics are incorporated as a ripening signal, influencing sugar uptake, respiration, hydraulic conductance and transpiration processes. The model has been primarily tested on blueberries, a fruit with well-characterised ABA responses. Simulations show predictive accuracy and explanatory capability for fruit mass under variable climatic conditions. Notably, the model effectively simulates the impacts of environmental stresses such as heat, cold and drought, capturing the resulting physiological delays in fruit growth. Our research underscores the potential of integrating phytohormonal responses into biophysical models, providing key insights into fruit growth dynamics and practical guidance for optimising crop management under increasing climate uncertainties.

## 1. Introduction

Global warming and extreme weather events are causing unpredictable patterns in fruit development (Fontúrbel et al., 2018; Houston et al., 2018; Vanalli et al., 2021), affecting both crop yields and production quality. These challenges pose significant concerns for natural ecosystems and agriculture, making an understanding of fruit development crucial for effective mitigation.

Fleshy fruit development proceeds through three stages: set, growth and ripening (Fenn & Giovannoni, 2021). After fertilisation, the fruit set stage leads to the growth stage involving cell division and expansion, determining fruit size and weight. Ripening then follows, evolving ripening qualities like flavour and colour. Throughout these stages, the fruit acts as a sink for carbon and water as a resource for fruit development (Ren et al., 2023). The growth and ripening stages proceed sequentially or simultaneously, depending on the species, genotypes and environmental conditions, regulating carbon and water allocations for optimal development.

Phytohormones orchestrate the stage of fruit growth and ripening (Fenn & Giovannoni, 2021; Kou et al., 2021; Ren et al., 2023). During fruit growth, auxin and gibberellic acid synergistically regulate cell division and expansion, as commonly observed in many fruits. Fruit ripening can be classified into two types based on the primary phytohormones: climacteric, including fruits like apples and peaches, where ethylene is significant and non-climacteric, such as strawberries, blueberries and grapes, associated with abscisic acid (ABA). However, recent evidence indicates varying requirements for ABA within the ripening physiology of both fruit types (Fenn & Giovannoni, 2021; Ren et al., 2023).

In fruit, ABA primarily originates from endogenous biosynthesis within the fruit itself (Chung et al., 2019; Zhang et al., 2009) and determines fruit size and ripening times (Liao et al., 2018). In apples, ABA influences the transcriptional regulation of genes involved in sugar and

water metabolism. For example, in apples, MdWRKY9 interacts with MdbZIP23 and MdbZIP46, key ABA signal transducers, to enhance the regulatory effect on the expression of MdSWEET9b, a sugar transporter (Zhang et al., 2023). In addition to influencing sugar flux, ABA can modify water flux by affecting the total tissue and apoplastic water-soluble calcium concentration in fruit (de Freitas et al., 2011). These biochemical regulations drive the biophysical processes determining the accumulation of fruit biomass (Liao et al., 2018; Wang et al., 2022; Zhang et al., 2023).

Process-based modelling elucidates the interplay between environmental conditions and internal mechanisms in fruit growth (Grisafi et al., 2021). Most models focus on carbon and water dynamics, offering insights into fruit dry mass and water mass (Chen et al., 2021; Fishman & Génard, 1998; Lescourret & Génard, 2005; Zhu et al., 2019). Fishman and Génard (1998) developed a biophysical model for peach fruit growth, which has since been adapted and expanded for various fruits, including tomatoes (Liu et al., 2007) and grapes (Zhu et al., 2019). Liu et al. (2007) incorporated sugar metabolism into the tomato model, thereby enhancing estimations of structural and non-structural carbons and improving the predictions of osmotic pressure and fruit weight. Zhu et al. (2019) refined the carbon sink and source dynamics, refining carbon inputs of a grape growth model. These models well-described the overall estimation of biomass, responding to external stimuli; however, they lack the incorporation of phase shifts driven by ripening signals, particularly like ABA which plays a crucial role in regulating fruit growth and ripening.

In this study, we present a process-based model that integrates the quantitative effects of ABA on the biophysical growth of blueberry (*Vaccinium* spp.) fruit. Given that blueberry fruit growth is influenced by endogenous ABA (Chung et al., 2019; Zifkin et al., 2012), the integrative model could effectively describe biomass accumulation driven by ABA-induced biophysical changes. We developed an empirical model of ABA accumulation in fruit and coupled it with a biophysical fruit growth model for blueberry. After calibration and validation with blueberry fruit data, the model was simulated under climate change scenarios to assess the effect of global warming on fruit growth. Furthermore, by applying the model to regional climate data, we evaluated the performance in different environmental conditions, including abiotic stresses, and correlated these findings with yield data. The modelling process was implemented using the Cropbox framework, which features a domain-specific language designed for crop modelling with reduced technical complexity (Yun & Kim, 2022).

## 2. Materials and methods

### 2.1. Plant materials

Ten-year-old 'Bluecrop' highbush blueberry (*V. corymbosum*) shrubs were grown at the experimental orchard of Seoul National University, Suwon (37° 17′ N, 127° 00′ E), Republic of Korea. In the 2015 growing season, five blocks were designed; within each block, two shrubs were randomly selected to assess fruit characteristics. Fruits on each shrub were chosen using a consistent set of criteria: visually assessed, comparable sunlight exposure, matching canopy positions and identical anthesis dates. Twenty fruits from each shrub were harvested at eight intervals from 10 days after anthesis (DAA) to 80 DAA every 10 days.

Fruits were investigated for their dry, water and fresh masses to parameterise the fruit growth model. The dry mass was measured by drying the fruits at 70°C until a constant weight was achieved.

The initial water mass was then calculated for each fruit based on the difference between its fresh and dry masses.

To determine the ABA concentration at each growth stage, five fruits per shrub were separately collected at the eight DAAs. These samples were immediately frozen in liquid nitrogen and stored at −80°C.

In each block, data from one shrub was selected for calibrating the fruit growth model based on the fruit characteristics; data from the remaining shrubs were used for model testing.

### 2.2. Determination of ABA concentration

ABA concentration of 'Bluecrop' blueberry fruits was determined at the eight growth stages, according to Oh et al. (2018). Freeze-dried samples were rehydrated, homogenised and subjected to a series of centrifugation and lyophilisation steps. The processed samples were then analysed for ABA content using a Triple TOF 5600 Q-TOF LC-MS/MS (AB Sciex, Foster City, CA, USA) coupled with an Ultimate 3000 RS HPLC system (Thermo Dionex, Waltham, MA, USA). Chromatographic separation was performed in a formic acid and acetonitrile gradient, and for ABA quantification, multiple reaction monitoring was employed in the transition from 263.1 to 153.1 m/z. The ABA concentration was expressed as $\mu$g mg$^{-1}$ of fruit dry weight with a threshold of 0.01 $\mu$g mg$^{-1}$ for detection.

### 2.3. The model: incorporation of ABA actions in fruit growth via carbon and water fluxes

To incorporate the effect of endogenous ABA on fruit growth, we modified the biophysical fruit model developed by Fishman and Génard (1998) (Figure 1). This model conceptualises the fruit as a large single cell, separated from the phloem and xylem by a composite membrane. The dry, water and fresh masses of the fruit are calculated based on the carbon and water fluxes across the membrane. These fluxes are governed by thermodynamic equations that account for the membrane's hydraulic conductivity, differences in hydraulic and osmotic pressures, and the membrane's solute impermeability.

In the integrative model, we posit that endogenous ABA affects fruit growth by regulating carbon and water fluxes, a premise supported by physiological and biochemical research (Gutiérrez et al., 2021; Jia et al., 2016; Kobashi et al., 2001; Ofosu-Anim et al., 1996; Trivedi et al., 2019; Wheeler et al., 2009). ABA accumulation is modelled based on heat units to reflect the correlation between fruit development and ABA concentrations (Chung et al., 2019), where developmental changes are linked to heat accumulation (Marra et al., 2002). Recognising the different quantitative effects of ABA on carbon and water fluxes, we adopt empirically derived different equations for each flux component, including sugar uptake, respiration, the membrane's hydraulic conductivity and the permeability of the fruit skin. The variables in this model are detailed in Supplementary Table S1.

During the growth of 'Bluecrop' blueberry fruit, the fresh mass is calculated as the sum of dry mass ($s$, mg) and water mass ($w$, mg). The seed mass is not considered in this calculation, as it accounts for only 0.80–1.08% of the total mass, a negligible amount, according to Strik and Vance (2019). The rate of dry mass accumulation ($ds/dt$, mg h$^{-1}$) is quantified as the net result of the sugar uptake from the phloem ($U_s$, mg h$^{-1}$) and the carbon lost to fruit respiration ($R_f$, mg h$^{-1}$) (Equation (1)), where $U_s$ consists of active uptake through the apoplastic pathway ($U_a$, mg h$^{-1}$), mass flow through the symplastic pathway and passive diffusion across the membrane

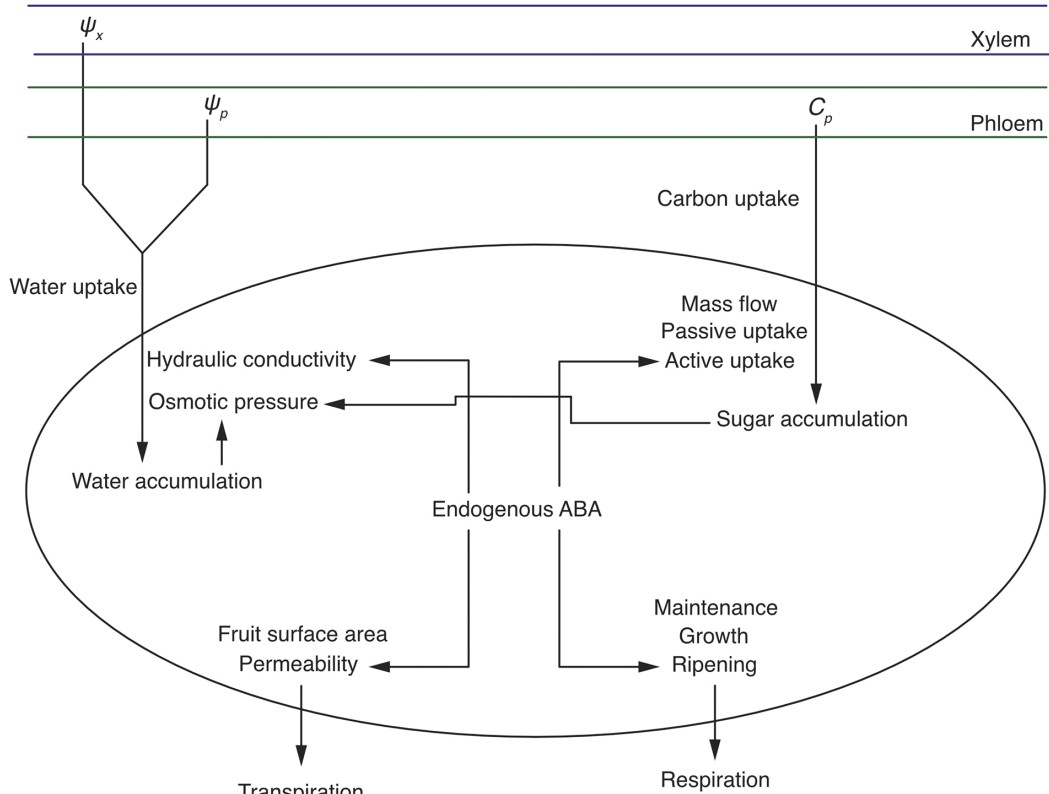

**Figure 1.** The schematic diagram for depicting the effect of endogenous abscisic acid (ABA) on the biophysical growth of fruit. Water enters the fruit (represented by the black-bordered oval) via the xylem and phloem, propelled by the water potential gradient between the xylem ($\Psi_x$) and phloem ($\Psi_p$), and the fruit itself. Water loss occurs through transpiration. Sugars from the phloem sap (sucrose concentration, $C_p$) are transferred into the fruit through mass flow, and passive and active uptake mechanisms, with a portion being respired. Endogenous ABA modulates biophysical growth by altering hydraulic conductivity and the permeability of the fruit skin during water flux; it also fine-tunes active sugar uptake and respiration associated with ripening during carbon flux.

(Fishman & Génard, 1998). The rate of water mass accumulation ($dw/dt$, mg h$^{-1}$) is determined by the total water uptake from the xylem ($U_x$, mg h$^{-1}$) and phloem ($U_p$, mg h$^{-1}$) against the water lost through fruit transpiration ($T_f$, mg h$^{-1}$) (Equation (2)):

$$\frac{ds}{dt} = U_s - R_f, \tag{1}$$

$$\frac{dw}{dt} = U_x + U_p - T_f. \tag{2}$$

The quantitative response of ABA is integrated into the model to predict the accumulation of dry and water masses by characterising ABA concentration. These concentrations initially surge and then slightly diminish during the late ripening stage in non-climacteric fruits, including blueberry (Watanabe et al., 2021; Zifkin et al., 2012), grape (Stacey et al., 2009; Villalobos-González et al., 2016) and strawberry fruits (Liao et al., 2018). Given the absence of previous attempts to model ABA accumulation patterns in line with experimental data, a beta growth equation is empirically implemented (Equation (3)), which is driven by cumulative growing degree hours (cGDH, K) during the entire day (Equation (4)). Subsequently, the ABA concentration ($ABA_{conc}$, ug mg$^{-1}$ of $s$) was normalised ($ABA_{norm}$) to be utilised for carbon and water fluxes (Equation (5)):

$$\mathrm{ABA_{conc}} = \mathrm{ABA}_f \frac{(\mathrm{ABA}_e - c\mathrm{GDH})}{(\mathrm{ABA}_e - \mathrm{ABA}_m)} \left(\frac{c\mathrm{GDH}}{\mathrm{ABA}_m}\right)^{\frac{\mathrm{ABA}_m}{(\mathrm{ABA}_e - \mathrm{ABA}_m)}}, \tag{3}$$

$$c\mathrm{GDH} = \sum \left(\min\left(T, T_0\right) - T_b\right), \tag{4}$$

$$\mathrm{ABA_{norm}} = \frac{\mathrm{ABA_{conc}} - \mathrm{ABA}_0}{\mathrm{ABA}_f - \mathrm{ABA}_0}, \tag{5}$$

where $ABA_0$ (ug mg$^{-1}$ of $s$) and $ABA_f$ (ug mg$^{-1}$ of $s$) are the initial and final ABA concentrations, respectively. $cGDH$ is calculated by summing the difference between the base temperature ($T_b$, K) and effective temperature, which equals an ambient temperature ($T$, K) capped by optimal temperature ($T_0$, K). $ABA_e$ represents an environmental factor affecting $cGDH$ as an equilibrium point; $ABA_m$ is the baseline value of $cGDH$ that triggers the accumulation of ABA. $ABA_{norm}$ (dimensionless) is applied to adjust active sugar uptakes, which is based on biological interaction where endogenous ABA in fruit affects enzyme activation, including hexose transporters (Murcia et al., 2016). Specifically, ABA enhances sugar uptake by activating relevant enzymes up to a certain concentration level, beyond which the rate of sugar uptake decreases:

$$\mathrm{ABA_{sugar}} = k_{\mathrm{ABA}} \frac{(\mathrm{STP_e} - \mathrm{ABA_{norm}})}{(\mathrm{STP_e} - \mathrm{STP_m})} \left(\frac{\mathrm{ABA_{norm}}}{\mathrm{STP_m}}\right)^{\frac{\mathrm{STP_m}}{(\mathrm{STP_e} - \mathrm{STP_m})}}, \tag{6}$$

$$U_a = \frac{\mathrm{ABA_{sugar}} v_m C_p}{(K_m + C_p)\left(1 + e^{\frac{-t^*}{\tau}}\right)}, \tag{7}$$

where $ABA_{sugar}$ (dimensionless) represents the ABA-modulated sugar uptake factor based on beta function with a scaling factor ($k_{ABA}$). $STP_e$ (dimensionless) and $STP_m$ (dimensionless) are thresholds defining the range within which ABA concentration optimally

enhances sugar uptake; $STP_e$ is the upper threshold beyond which the effectiveness of ABA starts to decline, and $STP_m$ is the minimum threshold below which the effect of ABA on sugar uptake is minimal. The $v_m$ represents the maximum rate of sugar transport. $K_m$ (dimensionless) is the Michaelis–Menten constant for sugar transport: $t*$ (h) and $\tau$ (h), describing the activity of an inhibitor.

In biophysical models, fruit respiration is traditionally partitioned into growth respiration and maintenance respiration (Fishman & Génard, 1998; Zhu et al., 2019); the growth respiration is proportional to d$s$/d$t$, and the maintenance respiration is proportional to $s$ (Thornley & Johnson, 1990). To more accurately reflect respiration changes based on fruit phenology, we introduce a novel 'ripening respiration' category. This category, describing respiration during metabolic shifts that evolve fruit qualities during the ripening phase (Perotti et al., 2014; Saltveit, 2019), is distinctly separate from the traditional concepts of maintenance and growth respiration. This addition is justified by observed changes in respiration patterns following ripening initiation across various fruit types, albeit to varying degrees (Kou et al., 2021). To integrate ripening respiration into the model, we leverage ABA concentration as a ripening signal, introducing a ripening coefficient ($q_r$, mg h$^{-1}$) that reflects the effect of ABA on respiratory dynamics. $R_f$, fruit respiration, is thus formulated as follows:

$$R_f = q_g \frac{ds}{dt} + q_m(T)s + q_r \text{ABA}_{\text{norm}}, \tag{8}$$

where $q_q$ (dimensionless) and $q_m(T)$ (h$^{-1}$) are the coefficients for growth and maintenance, respectively. The effect of air temperature ($T$) on maintenance respiration is addressed by the $Q_{10}$ concept, which quantifies the rate of increase in a biological process, such as respiration, for every 10°C rise in temperature (Fishman & Génard, 1998; Pavel & DeJong, 1993). With Equations (7) and (8), Equation (1) was calculated to estimate dry mass. The subsequent equations are followed by the berry growth module of Zhu et al. (2019).

Water flux in fruit represents the difference between total water uptake and transpirational water loss. Water uptake is determined by the hydraulic conductivity ($L$, mg cm$^{-2}$ MPa$^{-1}$ h$^{-1}$) of the phloem and xylem, which are assumed to have identical values (Fishman & Génard, 1998). We introduce incorporating the effect of ABA to reflect changes in $L$ attributable to alteration in fruit surface characteristics during the ripening phase (Peschel et al., 2003; Trivedi et al., 2019). The ABA effect on the conductivity is expressed through the following equation:

$$L = L_{\text{max}} e^{(-k_L ABA_{\text{norm}})}, \tag{9}$$

where $L_{\text{max}}$ (mg cm$^{-2}$ MPa$^{-1}$ h$^{-1}$) denotes the maximum conductivity achievable in the absence of ABA modulation. The constant $k_L$ determines the sensitivity of conductivity to ABA levels. The $L$ is implemented to $U_x$ and $U_p$ of Equation (2) to estimate $w$ with other parameters and followed equations described by Zhu et al. (2019). In addition, the equation for cell extensibility is employed to represent variations in the cell wall extensibility coefficient rather than treating it as a constant (Liu et al., 2007).

Fruit transpiration is assumed to be proportional to the fruit surface area ($A_f$, cm$^2$) and to be driven by the difference in relative humidity between the air-filled space within the fruit ($H_f$, dimensionless) and the ambient atmosphere ($H_a$, dimensionless):

$$T_f = A_f \alpha \rho (H_f - H_a). \tag{10}$$

In this equation, $\alpha$ (g cm$^{-3}$) is a temperature-dependent coefficient that converts the relative humidity gradient into a water flux term

(Fishman & Génard, 1998). $\rho$ (cm h$^{-1}$) is the solute permeability coefficient of the fruit skin, a crucial factor in determining the rate of transpiration. Equation (10) describes how the $\rho$ is affected by endogenous ABA:

$$\rho = \rho_{\text{min}} + \rho_0 e^{(-k_p ABA_{\text{norm}})}. \tag{11}$$

Here, $\rho_{\text{min}}$ (cm h$^{-1}$) represents the coefficient for minimum solute permeability of the fruit skin and $\rho_0$ (cm h$^{-1}$) is the scaling factor. The exponential term captures the effect of ABA on $p$, modulated by a constant $k_p$ (dimensionless). The integration indicates that as ABA increases, there is a corresponding decrease in $\rho$, reflecting the influence of ABA on enhancing the barrier properties of the fruit skin. This approach is based on ABA-induced changes in cell wall and cuticle properties, according to Curvers et al. (2010), leading to a reduction in water loss through the fruit skin.

After the carbon and water fluxes have been calculated, integration of Equations (1) and (2) over time yields the main state variables of the system $s(t)$ and $w(t)$ with each initial mass ($s_0$ and $w_0$, respectively).

$$s(t) = s_0 + \int (U_s - R_f) \, dt, \tag{12}$$

$$w(t) = w_0 + \int (U_x + U_p - T_f) \, dt. \tag{13}$$

### Model implementation, parameterisation and validation

The fruit growth model was comprehensively established within the Cropbox modelling framework (Yun & Kim, 2022), which enhances modelling efficiency and precision, allowing a refined focus on the biological processes that underpin fruit growth simulations.

Model parameterisation utilised both experimental data and literature. Hourly air temperature and relative humidity were captured using WatchDog 2450 data loggers (Spectrum Technologies, Inc., Aurora, IL, USA). Stem water potential was set to oscillate diurnally between −0.10 and −1.8 MPa (Glass et al., 2005), and the phloem sucrose concentration was set between 15 and 100 mM (Zhu et al., 2019).

The model encompasses 27 parameters: 13 derived through calibration and the remainder from literature (Table 1). Calibration focused on ABA accumulation, fruit surface transpiration, composite membrane area, sugar uptake and hydraulic conductance. Parameters for ABA accumulation were calibrated independently before integration into the full model, which was then further calibrated using fresh mass data. These parameters were utilised in model validation against a distinct dataset to assess performance.

### 2.4. Model goodness-of-fit analyses

The goodness-of-fit was assessed using observed data for ABA concentrations and fruit masses. The evaluation employed five metrics: mean absolute error (MAE), root mean square error (RMSE), normalised RMSE (NRMSE), the index of agreement ($d_r$) and the Nash–Sutcliffe model efficiency (EF):

$$\text{MAE} = \frac{1}{n} \sum_{i=1}^{n} |y_i - \widehat{y_i}|, \tag{14}$$

$$\text{RMSE} = \sqrt{\frac{1}{n} \sum_{i=1}^{n} (y_i - \widehat{y_i})^2}, \tag{15}$$

**Table 1.** List of symbols and the estimates of the model parameters used for 'Bluecrop' highbush blueberry (*Vaccinium corymbosum*)

| Symbol | Description | Value | Unit | Source |
|---|---|---|---|---|
| **Growing degree hours** | | | | |
| $T_b$ | Base temperature | 281.0 | K | Zheng et al. (2017) |
| $T_o$ | Optimal temperature | 305.6 | K | Zheng et al. (2017) |
| **Abscisic acid (ABA) accumulation** | | | | |
| $ABA_m$ | Baseline of *cGDH* | 18287 | K | Calibration |
| $ABA_e$ | Upper threshold of *cGDH* | 23683 | K | Calibration |
| **Fruit surface transpiration** | | | | |
| $\gamma$ | Empirical coefficient for fruit surface area | 4.24 | $cm^2\,g^{-1}$ | Jorquera-Fontena et al. (2017) |
| $\eta$ | Empirical coefficient for fruit surface area | 0.70 | Dimensionless | Jorquera-Fontena et al. (2017) |
| $\rho_{min}$ | Minimum fruit surface conductance to water vapour | 1.005 | $cm\,h^{-1}$ | Calibration |
| $\rho_0$ | Scaling factor | 193.401 | $cm\,h^{-1}$ | Calibration |
| $k_p$ | Exponential decay rate | 54 | Dimensionless | Calibration |
| **Composite membrane area** | | | | |
| $a$ | Coefficient for converting fruit surface area to membrane area | 9.13e−4 | Dimensionless | Calibration |
| **Sugar uptake** | | | | |
| $\sigma_p$ | Reflection coefficient for sugar for entering the composite membrane | 0.9 | Dimensionless | Fishman and Génard (1998) |
| $v_m$ | Maximal rate of active transport | 2.12e−3 | mg sucrose $(g\ of\ s)^{-1}\,h^{-1}$ | Calibration |
| $K_m$ | Michaelis constant for the equation of active transport | 0.08 | Dimensionless | Fishman and Génard (1998) |
| $p_s$ | Permeability of the composite membrane for sugar transport | 0.27 | $mg\,cm^{-2}\,h^{-1}$ | Fishman and Génard (1998) |
| $t*$ | kinetic parameter for inhibitor accumulation | 1138.8 | h | Fishman and Génard (1998) |
| $\tau$ | Characteristic time for inhibitor accumulation | 216.95 | h | Fishman and Génard (1998) |
| $\sigma_P$ | Reflection coefficient of the composite membrane for sugar | 0.9 | Dimensionless | Fishman and Génard (1998) |
| $k_{ABA}$ | Scaling factor | 4.95 | Dimensionless | Calibration |
| $STP_e$ | Upper threshold of normalised ABA concentration ($ABA_{norm}$) | 0.51 | Dimensionless | Calibration |
| $STP_m$ | Minimum threshold of $ABA_{norm}$ | 0.05 | Dimensionless | Calibration |
| **Hydraulic conductance** | | | | |
| $L_{max}$ | Conductivity of the composite membrane for water transport | 15 | $mg\,cm^{-2}\,MPa^{-1}\,h^{-1}$ | (Zhu et al., 2019) |
| $\phi_{max}$ | Cell wall extensibility coefficient in Lockhart's equation | 0.4 | $MPa^{-1}\,h^{-1}$ | Jorquera-Fontena et al. (2017) |
| $Y$ | Threshold value of hydrostatic pressure needed for fruit development | 2.47 | MPa | Calibration |
| $k_L$ | Coefficient for sensitivity of conductivity to ABA | 2.51 | Dimensionless | Calibration |
| **Respiration** | | | | |
| $q_g$ | Maintenance respiration coefficient | 0.21 | Dimensionless | Fishman and Génard (1998) |
| $Q_{10}$ | Temperature ratio of maintenance respiration | 2.03 | Dimensionless | Thornley and Cannell (2000) |
| $q_r$ | Coefficient for ripening respiration related to ABA | 0.01e−5 | $mg\,h^{-1}$ | Calibration |

$$\mathrm{NRMSE} = \frac{\mathrm{RMSE}}{y_{max} - y_{min}}, \tag{16}$$

$$d_r = 1 - \frac{\sum\limits_{i=1}^{n}\left(y_i - \widehat{y_i}\right)^2}{\sum\limits_{i=1}^{n}\left(|y_i - \overline{y}| + |\widehat{y_i} - \overline{y}|\right)^2}, \tag{17}$$

$$\mathrm{EF} = 1 - \frac{\sum_{i=1}^{n}\left(y_i - \widehat{y_i}\right)^2}{\sum_{i=1}^{n}\left(y_i - \overline{y}\right)^2}, \tag{18}$$

where $y_i$ denotes the observed value, $\hat{y}_i$ denotes the predicted value and $\overline{y}$ denotes the mean of observed values, with $n$ representing the total observations. The terms $y_{max}$ and $y_{min}$ indicate the maximum and minimum observed values, respectively. MAE and RMSE quantify the average prediction error magnitude, assessing the model prediction. NRMSE normalises RMSE against the observation data range, offering a unitless metric for variable comparison. The indices, $d_r$ and EF, approaching 1, signify excellent model fit and predictive accuracy.

## 2.5. Sensitivity analysis

A local sensitivity analysis was performed to identify the contribution of individual parameters to fresh mass, as per the methods described by Thornley et al. (1981). Thirteen parameters used for the calibration were grouped into six categories: ABA accumulation, composite membrane area, fruit surface transpiration, hydraulic conductance, respiration and sugar uptake. The sensitivity of each parameter was quantified using a sensitivity coefficient.

The coefficient was the relative change in fresh mass in response to a proportional adjustment in the parameter's value and expressed as a percentage:

$$\text{Sensitivity coefficient } (\%) = 10\frac{\Delta W/W}{\Delta P/P}, \qquad (19)$$

where $W$ represented the final berry fresh weight obtained under baseline calibrated parameters, and $P$ referred to the original calibrated parameter value. The term $\Delta W$ and $\Delta P$ indicated the changes in berry fresh weight and parameter value, respectively. The factor 10 normalises the coefficient to represent the percentage change in berry fresh weight corresponding to a standardised 10% increase in the parameter, with other parameters unchanged.

### 2.6. What-if scenario simulations

The robustness and versatility of the fruit growth model were assessed through what-if scenario simulations under various climates. The first scenario assessed the effects of temperature increases on blueberry growth, adjusting the baseline temperature by 3°C and 5°C for each hourly input while keeping relative humidity constant. A second scenario examined geographic growth variations using real climate data from Seattle, Washington, USA, sourced from AgWeatherNet, covering May 5 to August 31 from 2018 to 2023. Additionally, variations in anthesis dates were simulated within this context to evaluate their impact on fruit growth, thus testing the model's relevance to practical agricultural scenarios.

## 3. Results

### 3.1. Calibration and evaluation of the fruit growth model

The calibration of the fruit growth model delineated the patterns of ABA accumulation and the variations in fruit masses of blueberry fruit (Figure 2). Initially, the ABA concentration increased gradually, then surged rapidly to a peak of 32.30 $\mu$g g$^{-1}$, and ultimately declined swiftly (Figure 2a). The MAE and RMSE values were 2.03 and 2.83 mg fruit$^{-1}$, respectively (Table 2), less precise fit due to overestimation of the initial rapid ascent and underestimation thereafter (Figure 2a). Despite this, the low NRMSE (0.26) and high $d_r$ (0.91) indicate that the estimation of the model represented the overall trends in the pattern of ABA accumulation.

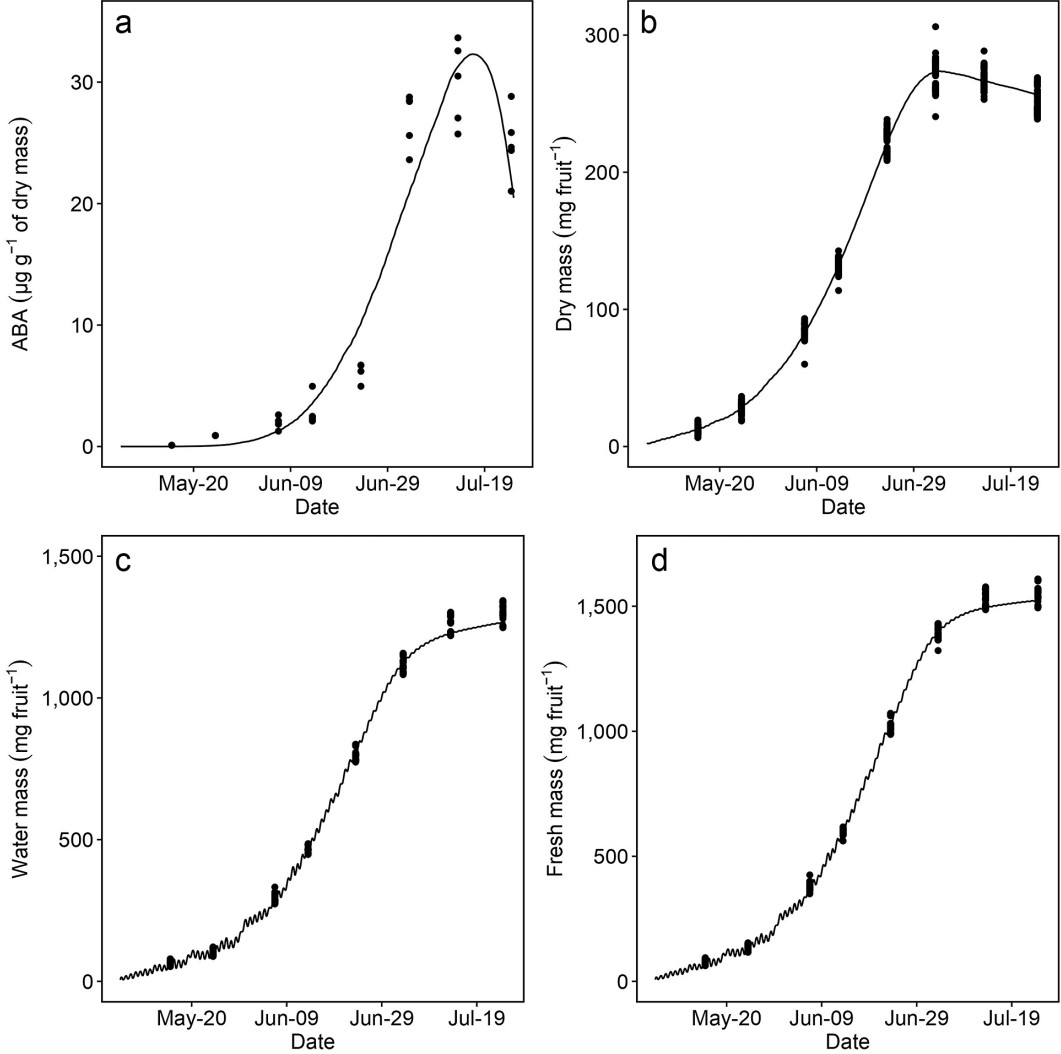

**Figure 2.** Comparison of simulations (lines) and measurements (points) of the concentration of abscisic acid (mg g$^{-1}$ of dry mass) (a) dry mass (mg fruit$^{-1}$) (b) water mass (mg fruit$^{-1}$) (c) and fresh mass (mg fruit$^{-1}$) (d) of 'Bluecrop' blueberry (*Vaccinium corymbosum*) fruit.

**Table 2.** Mean absolute error (MAE), root mean square error (RMSE) and normalised root mean square error (NRMSE), the index of agreement ($d_r$) and efficiency coefficient (EF) between the observed and simulated the concentration of abscisic acid (ABA), dry mass, water mass and fresh mass, for 'Bluecrop' blueberry (*Vaccinium corymbosum*) fruit

| Goodness-of-fit | Variable | Unit | Calibration | Validation |
|---|---|---|---|---|
| MAE | ABA | $\mu$mol g$^{-1}$ DW | 2.03 | 2.31 |
| | Dry mass | mg fruit$^{-1}$ | 7.79 | 8.93 |
| | Water mass | mg fruit$^{-1}$ | 27.40 | 29.48 |
| | Total mass | mg fruit$^{-1}$ | 26.26 | 28.16 |
| RMSE | ABA | $\mu$mol g$^{-1}$ DW | 2.82 | 3.26 |
| | Dry mass | mg fruit$^{-1}$ | 10.89 | 12.76 |
| | Water mass | mg fruit$^{-1}$ | 38.13 | 41.53 |
| | Total mass | mg fruit$^{-1}$ | 37.80 | 40.14 |
| NRMSE | ABA | – | 0.26 | 0.28 |
| | Dry mass | – | 0.07 | 0.08 |
| | Water mass | – | 0.06 | 0.06 |
| | Total mass | – | 0.05 | 0.05 |
| $d_r$ | ABA | – | 0.91 | 0.90 |
| | Dry mass | – | 0.96 | 0.94 |
| | Water mass | – | 0.97 | 0.97 |
| | Total mass | – | 0.98 | 0.97 |
| EF | ABA | – | 0.94 | 0.93 |
| | Dry mass | – | 0.99 | 0.98 |
| | Water mass | – | 0.99 | 0.99 |
| | Total mass | – | 0.99 | 0.99 |

The model effectively represented dry, water and fresh masses (Figure 2b–d). The dry mass exhibited a rapid rise followed by a slight decline in the later stages of fruit growth (Figure 2b). This pattern was aligned with the differences between the sugar uptake and respiration (Supplementary Figure S1A,B). Notably, the decline in dry mass was predominantly due to a more pronounced decrease in the sugar uptake compared to the respiration. The water mass exhibited a rapid increase akin to the dry mass, but then transitioned into a slower growth phase (Figure 2b), without the decline observed in the dry mass (Figure 2c). The trend was consistent with the dynamics of water uptake and transpiration (Supplementary Figure S1C,D); water uptake exceeded the transpiration in most growth stages. The pattern of the fresh mass (Figure 2d) closely mirrored that of the water mass (Figure 2c), which is attributable to the water mass constituting a larger proportion of the fresh mass compared to the dry mass (Figure 2b,c). The estimations for dry, water and fresh masses were more robust than those for ABA (Figure 2), with low variance, as indicated by the values of RMSE and MAE (Table 2). Additionally, the values for NRMSE (0.05–0.07), $d_r$ (0.96–0.98) and EF (0.94–0.99) indicate higher precisions for fruit masses compared to ABA.

The validation of the fruit growth model exhibited trends similar to those observed during calibration for both ABA accumulation and the estimation of fruit masses (Figure 3, Table 2); the pattern of overestimation and underestimation in ABA accumulation during validation (Figure 3a) mirrored the trends seen in the calibration (Figure 2a). While the variance in goodness-of-fit measures slightly increased in the validation (Table 2), the values still indicate a well-fitted model to the observed data. For example, the $d_r$ values during validation ranged from 0.90 to 0.97.

### 3.2. Parameter sensitivity in fruit masses

In the sensitivity analysis, where parameters were increased by 10%, a range of changes in fruit masses was observed, spanning from −7.15% to 24.08% (Figure 4). The study encompassed six categories, wherein parameters linked to ABA accumulation, fruit surface transpiration and sugar uptake impacted fruit masses both positively and negatively, exhibiting consistent trends across all mass types. The parameter of the composite membrane area category, however, had a decrease in dry mass alongside an increase in water and fresh masses.

In the respiration category, represented by the $q_r$, no noticeable influence on fruit masses was detected (Figure 4). In the sugar uptake category, the parameters $k_{ABA}$ and $v_m$ stood out, registering significant increments in fruit masses (17.58%–24.08% for $k_{ABA}$ and 16.14%–22.04% for $v_m$), thereby marking them as the most impactful parameters. Other parameters led to changes in fruit mass below the 10% threshold.

### 3.3. Effect of temperature increase on fruit growth

Simulations for the effect of rising average temperatures indicate changes in ABA concentration patterns and fruit masses (Figure 5). With higher temperatures, the onset of ABA accumulation occurred earlier, and peak concentrations were reached more rapidly (Figure 5a). This effect was attributed to increases in *cGDH*, as all other variables and parameters remained constant, isolating temperature as the primary variable. Despite these temperature-induced shifts, the initial and final ABA concentrations, predefined constants in the fruit growth model, remained unaffected by climatic changes.

With rising temperatures, a synchronous pattern emerged in the growth rates and final weights of fruit masses (Figure 5b–d). The dry mass exhibited a quicker increase at elevated temperatures, achieving its maximum earlier (Figure 5b). However, the peak mass was lower in scenarios with higher temperatures. Post-peak, simulations for dry mass showed declining trends, while the water mass increased slightly (Figure 5c). The fresh mass, predominantly influenced by the water mass, followed a similar trend to the water mass rather than the dry mass (Figure 5d).

Throughout the simulations, the variations in ABA concentrations and fruit masses (Figure 5) were associated with the dynamics of carbon and water fluxes. The impact was pronounced, especially between June 15 and June 30 (Supplementary Figure S2). In this period, under the actual climate condition, dry mass continued to rise (Figure 5b) due to the sugar uptake surpassing respiration (Supplementary Figure S3A,B), despite a minor reduction in sugar uptake. Consequently, the carbon flux remained positive (Supplementary Figure S3C). Conversely, in scenarios with temperature increases of 3°C and 5°C, there was a marked reduction in sugar uptake and a slight decrease in respiration, leading to a net negative carbon flux (Supplementary Figure S2B,C). These conditions led to the dry mass peaking earlier and then declining (Figure 5b).

In parallel, the water mass changes (Figure 5c) were closely aligned with the interplay between water uptake and transpiration

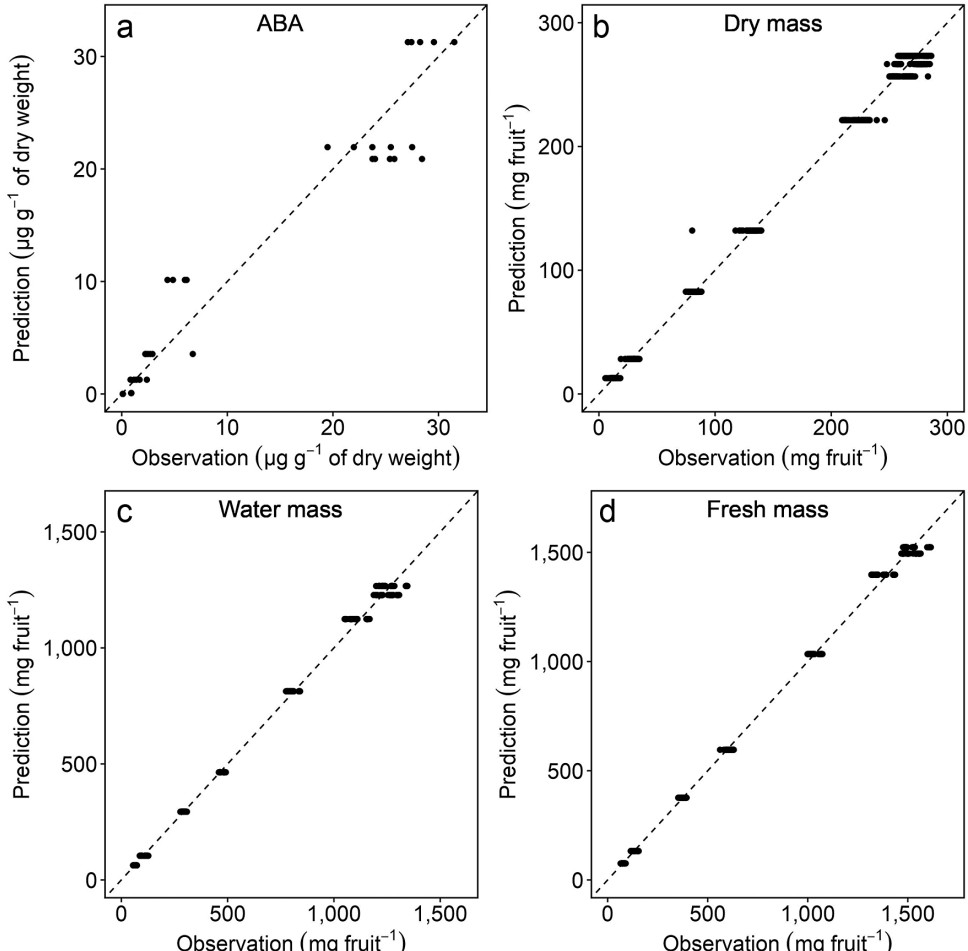

**Figure 3.** Validation of the fruit growth model by comparison of predicted and observed values of the concentration of abscisic acid (mg g$^{-1}$ of dry mass) (a) dry mass (mg fruit$^{-1}$) (b) water mass (mg fruit$^{-1}$) (c) and fresh mass (mg fruit$^{-1}$) (D) of 'Bluecrop' blueberry (*Vaccinium corymbosum*) fruit.

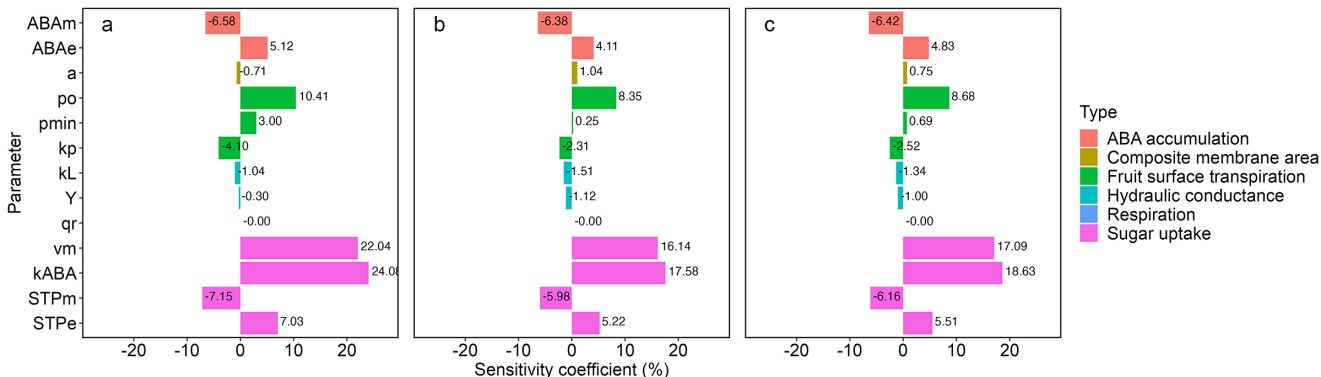

**Figure 4.** Sensitivity coefficients (coloured bars) calculated for dry mass (a), water mass (b) and fresh mass (c) to variations in calibrated parameters for 'Bluecrop' blueberry (*Vaccinium corymbosum*) fruit. The default value of a parameter, as noted in Table 1, was changed to 10%, while all other parameters were kept at their default values during the sensitivity analysis. $ABA_m$, the baseline of cumulative growing degree hours ($cGDH$) for abscisic acid (ABA) accumulation; $ABA_e$, the upper threshold for $cGDH$ for ABA accumulation; $a$, coefficient for converting fruit surface area to membrane area; $p_0$, scaling factor for the solute permeability of the fruit skin; $p_{min}$, the coefficient for minimum solute permeability of the fruit skin, $k_p$, exponential decay rate for fruit surface transpiration; $k_L$, the coefficient for the sensitivity of conductivity to ABA; $Y$, the threshold value of hydrostatic pressure needed for the fruit growth; $v_m$, maximal rate of active sugar transport; $k_{ABA}$, scaling factor for active sugar uptake; $STP_m$, the upper threshold of normalised ABA concentration ($ABA_{nrom}$) for active sugar uptake; $STP_e$, the minimum threshold of $ABA_{norm}$ for active sugar uptake.

(Supplementary Figure S2D,E). Across all simulations, both these processes exhibited a gradual decline from June 15 to June 30. This decline was more pronounced at higher temperatures, primarily because the decrease in water uptake was more substantial than that in transpiration (Supplementary Figure S2E,F).

### 3.4. Simulation of the fruit growth model under various climatic conditions

The fruit growth model was simulated using regional climatic data from Seattle, WA, USA, spanning the years 2018–2023 (Figures 6

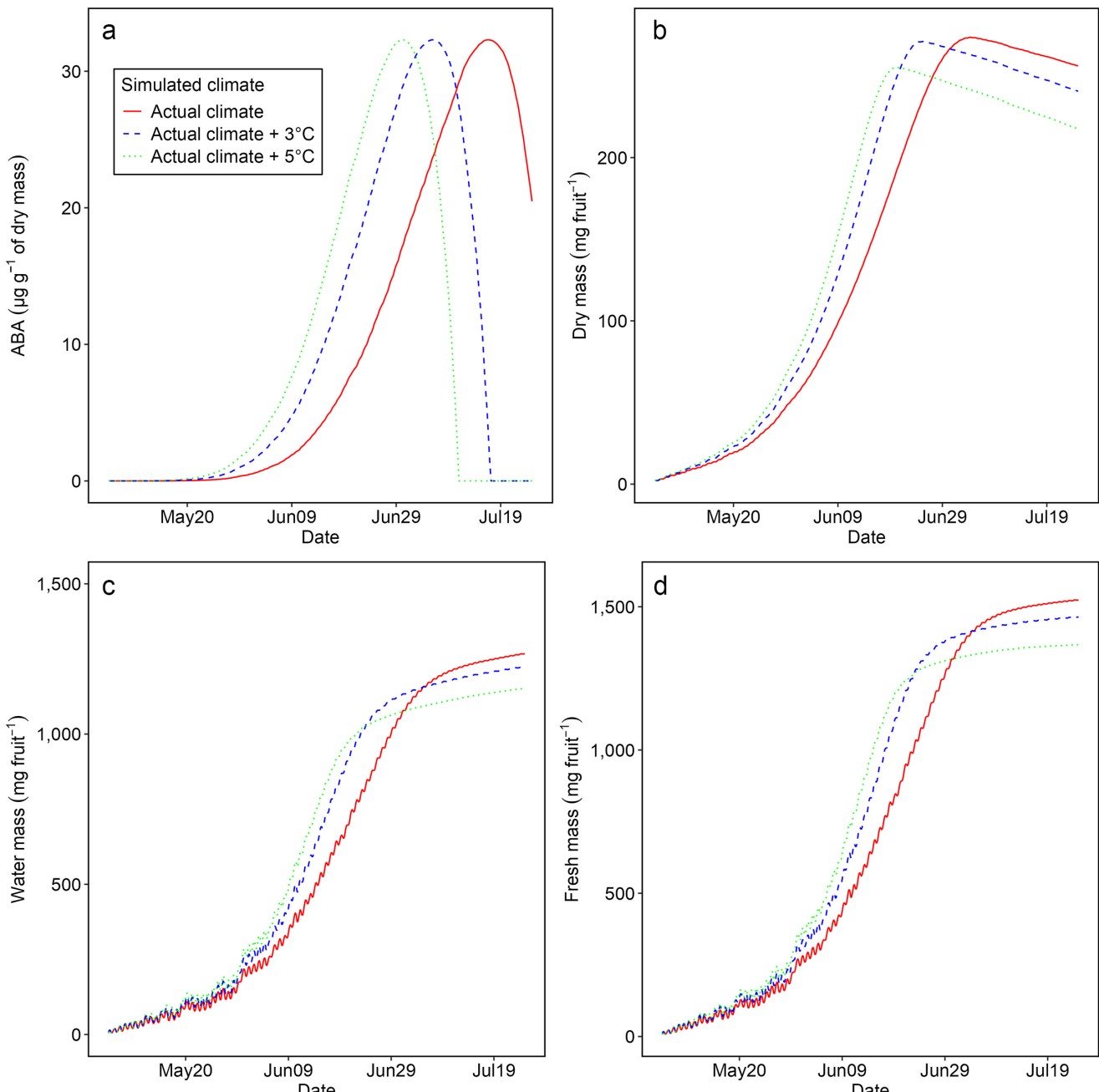

**Figure 5.** Simulations for the effect of increased average temperature on the concentration of abscisic acid (mg g$^{-1}$ of dry mass) (a), dry mass (mg fruit$^{-1}$) (b), water mass (mg fruit$^{-1}$) (c) and fresh mass (mg fruit$^{-1}$) (d) of 'Bluecrop' blueberry (*Vaccinium corymbosum*) fruit. The actual climate data (red solid lines) represent hourly air temperature and relative temperature from the 2015 season in Suwon, Republic of Korea. For increased temperatures, the hourly temperatures were augmented by +3°C (blue dashed lines) and +5°C (green dotted lines), while relative humidity was maintained consistent across all temperature scenarios.

and 7). Changes in fruit masses were compared during the growth stage and the potential harvest period with uniformly setting the start of anthesis to May 5th (Figure 6). The growth stage exhibited similar patterns to those previously observed in Figures 2 and 4; in the harvest period, dry mass consistently declined, while water and fresh masses plateaued during the harvest period.

Year-over-year analysis of fruit masses revealed 2023 as the year with the highest mass, followed by 2019, 2018, 2020, 2021 and 2022 (Figure 5a–c). The year 2022 stood out for having the lowest fruit masses across all periods, with fresh mass during the harvest period ranging from 614.45 mg to 655.24 mg, lower than the >1,000 mg

observed in other years (Figure 5c). This reduction in 2022 was associated with lower *cGDH* values (Figure 6d), attributable to reduced temperatures in the early fruit growth phase compared to the other years (Supplementary Figure S3).

Fruit growth in 2021 and 2022 was notably influenced by abrupt climatic changes. Specifically, between 26 and 29 June 2021, a marked decrease in both water and fresh masses was recorded (Figure 6b,c), coinciding with a temperature spike reaching 43.2°C (Supplementary Figure S3D) and a drastic drop in relative humidity to 24% (Supplementary Figure S4A). These conditions resulted in reduced water flux (Supplementary Figure S4D); however, a sharp

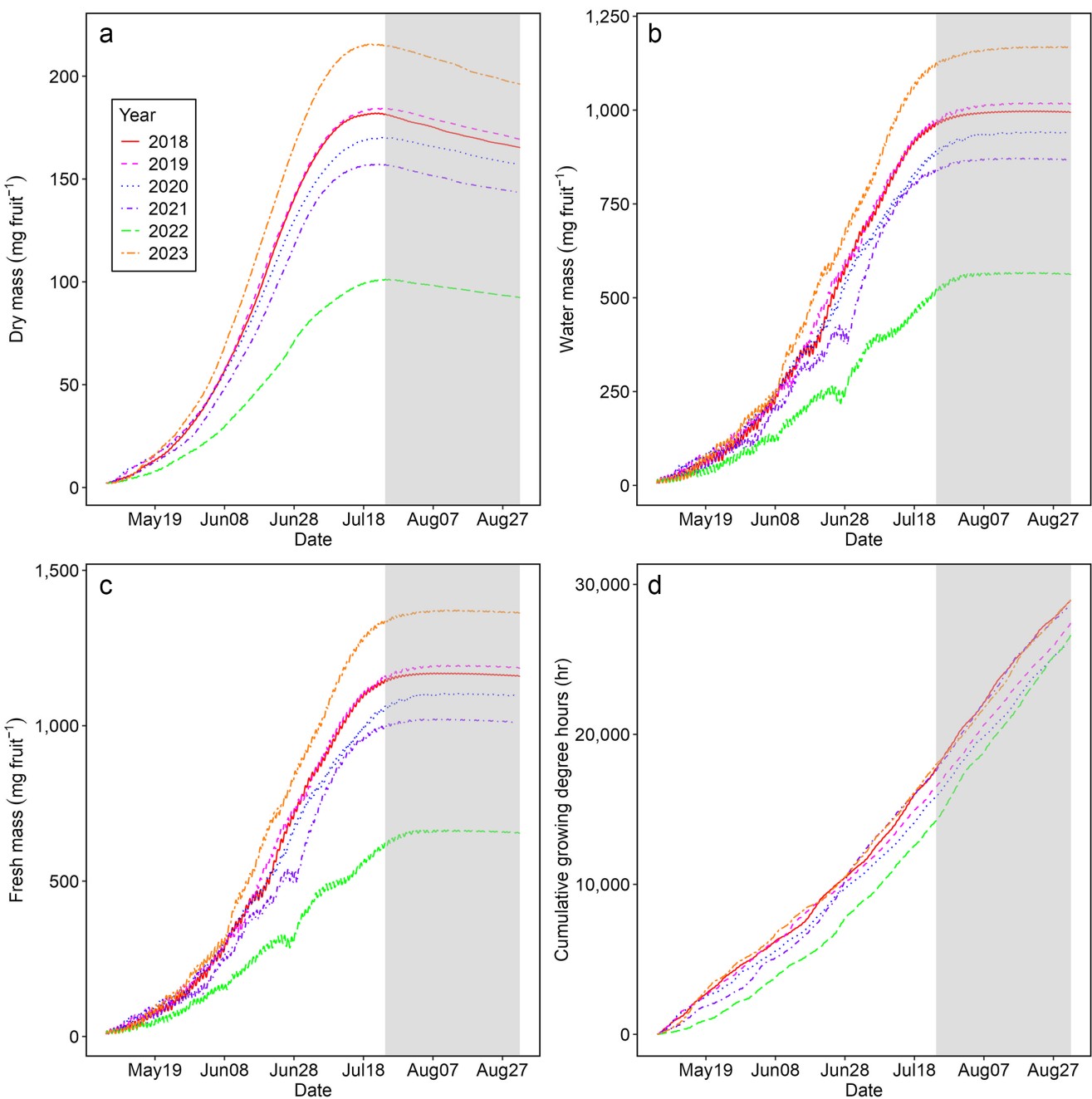

**Figure 6.** Simulations for the effect of yearly climatic conditions on dry mass (mg fruit⁻¹) (a), water mass (mg fruit⁻¹) (b) and fresh mass (mg fruit⁻¹) (c) of 'Bluecrop' blueberry (*Vaccinium corymbosum*) fruit, along with cumulative growing degree hours (h) (d). Hourly temperatures and relative humidity for Seattle, USA, were retrieved from AgWeatherNet (weather.wsu.edu): 2018, red solid lines; 2019, pink dashed lines; 2020, blue dotted lines; 2021, purple dotted-dashed lines; 2022, green long-dashed lines and 2023, orange two-dashed lines. The grey area was assumed as the harvest period from July 25 to August 31.

decrease in the solute permeability of the fruit skin mitigated water loss. A similar pattern was observed in 2022, characterised by high temperatures (Supplementary Figure S3E) and low relative humidity (Supplementary Figure S4A).

The simulated fresh masses were compared with the actual blueberry yield data from Washington state, USA (Figure 7) for heuristic evaluation of the model behaviour. The comparative analysis covered the years 2018–2022, noting that yield data for 2023 was unavailable. Interestingly, the trends observed in the simulated fresh mass closely matched the actual yield trends for these years.

### 3.5. The effect of anthesis timing on fruit growth

The timing of anthesis affected both the fresh mass (Figure 8) and the growth patterns of the fruit (Supplementary Figure S5). Under the simulation, variations in fresh mass across different years were with the standard deviations from 44.41 in 2018 to 188.29 in 2021, followed by 2018 (44.41), 2019 (72.78), 2020 (91.19), 2023 (93.10), 2022 (115.93), 2021 (188.29) (Figure 8). The years 2021 and 2022, which experienced abrupt climatic changes (Figure 6 and Supplementary Figure S3), showed higher variances in fresh mass. In these specific years, delaying the anthesis date proved beneficial,

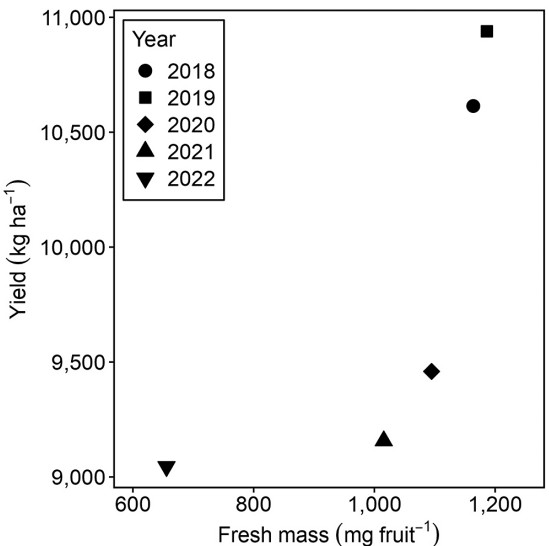

**Figure 7.** Comparison between the estimated fresh mass of blueberries and the actual blueberry fruit yield in Washington State, USA, from 2018 to 2022 (quickstats.nass. usda.gov). The fresh mass was simulated starting May 5, assuming this as the anthesis date, and calculated the mean value during the harvest period from July 24 to August 31.

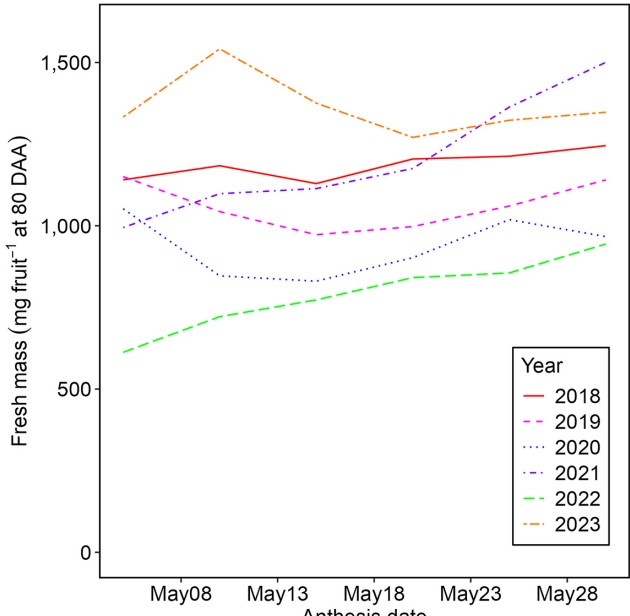

**Figure 8.** Variations in the fresh mass of 'Bluecrop' blueberry (*Vaccinium corymbosum*) fruit at 80 days after anthesis over a range of anthesis dates from the years 2018–2023 for Seattle, USA. Hourly temperatures and relative humidity were sourced from AgWeatherNet (weather.wsu.edu): 2018, red solid lines; 2019, pink dashed lines; 2020, blue dotted lines; 2021, purple dotted-dashed lines; 2022, green long-dashed lines and 2023, orange two-dashed lines.

mitigating the adverse effects of harsh environmental conditions and subsequently enhancing fruit growth. This adjustment led to significant increases in fresh mass, rising approximately 50% from 995 to 1,500 mg in 2021 and from 612 to 944 mg in 2022. However, despite these improvements in 2022, the coldest season (Figure 6d and Supplementary Figure S3E) consistently exhibited the lowest fresh mass across all anthesis dates.

The growth patterns of fresh mass were simulated across three distinct anthesis start dates: May 5, May 20 and June 1 (Supplementary Figure S5). All simulations exhibited a uniform growth pattern: an initial gradual increase, a swift expansion period, and then stabilisation. However, the timing of these phases and the ultimate fresh mass differed across scenarios. In the years 2018, 2021 and 2022, initiating anthesis earlier (May 5) led to a faster transition to the slow growth phase. In contrast, a later anthesis start (June 1) resulted in a more substantial increase in fresh mass (Supplementary Figure S5A,D,E). Particularly in 2021, a later anthesis date prolonged the rapid growth phase. In other seasons, fruits that began growth on May 5 or June 1 exhibited higher overall growth, whereas those with a May 20 start consistently showed the lowest growth levels (Figure 8).

## 4. Discussion

This study incorporated the quantitative ABA effect into the biophysical model, tailoring it for blueberry fruit, with potential applicability across a broad range of fruit species. The integrative model is a novel approach to depict fruit growth driven by a phytohormone, specifically focusing on ABA as a ripening signal to shift fruit growth phase (Chung et al., 2019; Fenn & Giovannoni, 2021; Jia et al., 2013). The model encapsulates the regulatory role of ABA for carbon and water uptake, respiration and transpiration, aligning with established knowledge of endogenous ABA roles. The simulation with environmental conditions and physiological responses closely mirrored empirical observations during the fruit growth (Figures 2 and 3). The sensitivity analysis highlighted the substantial impact of ABA, particularly in the modulation of sugar uptake associated with carbon flux (Figure 4).

### 4.1. Carbon flux during fruit growth

Dry mass in fruit growth is governed by carbon flux, with sugars derived predominantly from leaves contributing to mass gain, and respiration, resulting in carbon loss (Pavel & DeJong, 1993). This concept has been extensively explored in various biophysical models of fruit growth (Chen et al., 2021; Fishman & Génard, 1998; Liu et al., 2007; Zhou et al., 2023; Zhu et al., 2019). In these models, sugar uptake consists of active and passive mechanisms, as well as mass flow, which is governed by the sugar concentration gradient between the phloem and the fruit, assuming no reverse sugar flow the from fruit to the phloem. While passive sugar uptake and mass flow are primarily driven by biophysical processes, active sugar uptake involves a biochemical approach that typically describes enzymatic activities using parameters like $K_m$ and $v_m$ (Fishman & Génard, 1998). This approach adeptly captured the increase in dry mass during fruit growth, but there is an opportunity for further refinement to precisely estimate the decrease in dry mass often observed in the later stages of growth across a variety of fruit species.

Our model captured the dynamic pattern of dry mass (Figures 2b, 5b, 6a) by incorporating ABA response into active sugar uptake (Equations (6) and (7)) where $v_m$ was regulated by ABA concentrations, featuring minimum and upper thresholds. This change allowed our model to show decreasing dry mass in the later stages of fruit growth (Figure 2a), which aligns with a known observation that $v_m$ changes in strawberry fruit during the growth with no changes in $K_m$ (Ofosu-Anim et al., 1996). Dynamics of the active sugar uptake would be associated with sugar transporters during

fruit growth (Ren et al., 2023), with the transcriptional regulation (Li et al., 2022; Zhang et al., 2023). In apples, ABA regulated the expression of gene regulating tonoplast monosaccharide sugar transporter (*MdTMT1*) and sucrose transport (*MdSUT2*) by direct binding with the activity of a gene of ABA-signalling transcription factor, *MdAREB2*. A similar regulation was also observed in fruits of grapes (Wang et al., 2022), peaches (Kobashi et al., 2001) and strawberries (Jia et al., 2016), in which ABA regulated the expression of hexose transporters through ABA-signalling transcription factors.

Carbon loss in the model is quantified through respiration, encompassing growth, maintenance and ripening aspects. The respiration formula (Equation (8)) is newly designed to capture the dynamics of mitochondrial energy and reactive species metabolism, essential for developing fruit qualities such as colour, aroma and firmness during ripening (Kan et al., 2010; Perotti et al., 2014). According to Saltveit (2019), the oxidation of glucose, organic acid and fatty acid during ripening leads to $CO_2$ production. Physiological and biochemical research on ripening respiration has predominantly focused on climacteric fruits, characterised by a distinct peak in respiration coinciding with ethylene production during ripening. Although non-climacteric do not exhibit a marked increase in respiration, previous studies have reported that the application of exogenous ABA intensified the endogenous ABA synthesis of fruit, leading to increased ripening and fruit respiration in various fruits, including grapes (Wheeler et al., 2009), strawberries (Jia et al., 2013) and litchi (Lai et al., 2014). While our model incorporated the ripening respiration responding to ABA concentrations, the coefficient limited sensitivity to the fruit mass (Figure 3). The precise reason for this minimal effect, whether attributable to the characteristics of the non-climacteric fruit or the need for further refinement of ripening respiration, remains to be determined. To enhance the accuracy of the model, future developments should focus on incorporating enzymatic activities associated with ripening to provide a comprehensive view of ABA effects on carbon flux during fruit growth.

## 4.2. Water flux during fruit growth

Fruit masses are largely influenced by water flux, reflecting the high proportion of water in fresh fruits. Water flux depends on biophysical factors, including water potential gradient, hydraulic conductance and fruit surface permeability. To replicate the water flux, previous models have evolved in their approach to parameters related to water uptake and transpiration. Initially, parameters were considered constant (Chen et al., 2021; Fishman & Génard, 1998) and have been modified to be responsive to fresh mass (Zhu et al., 2019), to reflect observations of biophysical properties during physiological experiments. Our model implemented the ABA modulation of biophysical parameters governing water flux to align their changes associated with ripening (Gutiérrez et al., 2021; Trivedi et al., 2019). ABA modestly affected hydraulic conductance for water uptake (Figure 3), but significantly decreases the permeability, reducing surface conductance to water vapour and thereby limiting transpiration water loss.

Blueberry fruit exhibited a decrease in transpiration during fruit growth at a certain stage (Supplementary Figure S2), which would be associated with the growth of cuticular wax on the fruit surface during the ripening (Chu et al., 2018); blueberry fruit wax layer, primarily composed of aliphatic compounds, acts as a transpiration barrier (Vogg et al., 2004). A linkage between ABA and cuticular wax biosynthesis was demonstrated in various fruits

(Trivedi et al., 2019). In cherries, exogenous ABA accumulated wax compounds, particularly long-chain alkanes, which enhance the water impermeability (Gutiérrez et al., 2021). This effect was prominently demonstrated in our simulation of fruit growth, particularly in response to the abrupt high temperatures experienced in 2022 (Figure 6 and Supplementary Figure S4B). However, it remains unclear whether the ABA, as a ripening signal, also contributes to heat stress defence by reducing transpiration, or if additional ABA accumulates to protect the fruits from heat stress. Differentiating between the function of ripening and its stress-protective mechanisms is required for a precise depiction of its response.

## 4.3. Model responses to various climatic conditions

The fruit growth model adeptly responded to a wide range of climatic conditions (Figures 5, 6, 8). In fruit, the impact of temperature has been studied, manifesting in metabolites related to ripening, including sugar (Parker et al., 2020; Sadras & Petrie, 2012) and lycopene (Brandt et al., 2006); increased temperatures enhanced their accumulation, potentially indicating an acceleration of ripening. Our model demonstrated that the final phase of changes in fruit mass occurs earlier under scenarios with rising average temperatures (Figure 5). Considering the ABA role as a ripening signal that initiates changes in these metabolites (Fenn & Giovannoni, 2021), the model results (Figure 5) aligned with the ripening acceleration of the metabolite studies (Parker et al., 2020, Sadras & Petrie, 2012). This finding suggests that the fruit growth model can be expanded for modelling changes in fruit qualities associated with ABA during fruit growth and ripening.

Our model exhibited that rising temperatures led to a reduction in fresh mass (Figure 5b–d), a trend also observed in strawberries (Lobell & Field, 2011; Menzel, 2021), bananas (Varma & Bebber, 2019) and peaches (Sikhandakasmita et al., 2022). During the 2022 simulations using regional data, lower temperatures were linked to a decrease in the fresh mass of blueberries (Supplementary Figure S5E). This pattern of reduced yield due to lower temperatures was also evident in data from Washington state, USA (Figure 7). While our fruit growth model does not directly parallel the yield of blueberries, it indicates that lower temperatures negatively influence the mass of blueberry fruits. The variability in temperature effects among different genotypes (Wang & Camp, 2000; Zhang et al., 2022) underscores the need for further research to confirm these findings in field observations. Therefore, the model suggests that temperature fluctuations can cause changes in both the cultivation period and the fresh mass of fruits.

The anthesis timing influences blueberry fruit growth with yearly variations in pollination periods and bloom initiation. Strik and Vance (2019) reported that in Oregon, USA, in 2009, a 5% bloom of blueberries began around April 15, with full bloom by May 15, whereas in 2010, a 5% bloom started around March 25, with full bloom by May 5. These variations not only affect cultivation practices, with multiple harvest times, but also influence interaction with pollinators and whole-plant physiology, leading to variations in fruit qualities. Therefore, our model has the potential to enhance cultivation strategies for optimising fruit yield in response to climatic conditions.

## 4.4. Conclusions and future directions

The presented fruit growth model, developed within the Cropbox modelling framework, captures the dynamics of fruit development

by integrating biophysical growth with endogenous ABA within the fruit. By leveraging the capabilities of Cropbox, we successfully simulated fruit growth across different climates, replicating observed growth patterns and highlighting ABA's regulatory role in ripening. This integration offers valuable insights into the effects of various environmental conditions and helps assess the potential impacts of climate change on fruit development. Our approach also enables plant physiologists to focus on high-level abstractions of variable relationships and system structures, allowing them to expand and adapt the fruit model without dealing with low-level implementation details. However, the current model empirically estimates endogenous ABA accumulation, lacking a process-based approach for its biosynthesis, degradation and transport. The modular and flexible nature of the Cropbox framework provides an excellent platform for future enhancements. To extend the applicability of the model for evaluating various environmental effects and fruit qualities linked to ABA, future improvements should focus on modelling ABA dynamics based on the enzymatic reactions involved in its metabolism. Incorporating ABA transport via phloem and xylem, particularly from leaves in response to water stress, as demonstrated in recent studies (McAdam et al., 2016), and integrating source-sink dynamics would enhance model accuracy and robustness. Moreover, integrating the fruit module with a whole-plant hydraulic and carbon-allocation framework – thus capturing the upstream supply of water and assimilates – would substantially deepen mechanistic insight and strengthen the practical applicability. Despite these limitations, the blueberry model contributes to both academic research and agricultural practices, aiding in the development of effective strategies for crop management and improvement. For example, the model can help optimise fruit size to meet consumer preferences and inform precise temperature thresholds for deploying agricultural practices such as shading or automated climate-control systems. By offering a detailed understanding of the interplay between hormonal regulation and biophysical growth, the model serves as a valuable tool for predicting how fruits may respond to future climatic scenarios, ultimately supporting efforts to ensure food security amidst global environmental changes.

**Open peer review.** To view the open peer review materials for this article, please visit http://doi.org/10.1017/qpb.2025.10011.

**Supplementary material.** The supplementary material for this article can be found at http://doi.org/10.1017/qpb.2025.10011.

**Data availability statement.** The source code with the simulations within this manuscript is available via an open-access repository Zenodo (https://doi.org/10.5281/zenodo.10702539).

## Acknowledgements

The authors would like to thank Prof. Hee Jae Lee of Seoul National University for providing the blueberry orchards essential for their research. The authors would also like to appreciate the Plant Ecophysiology and Modelling Lab (PEML) team at University of Washington for their constructive feedback.

**Author contributions.** S.W.C. and S.-H.K conceived and designed the study. S.W.C. performed the formal analysis. S.W.C. and K.Y. constructed the model and wrote the code. S.W.C. and S.-H.K. undertook the model testing and refinement. S.W.C. wrote the first draft. S.W.C., K.Y. and S.-H.K. contributed to editing and revising the paper.

**Funding statement.** This work was in part supported by the Cooperative Research Program for Agricultural Science and Technology Development, Rural Development Administration, the Republic of Korea, under Grant Number (PJ015124012023).

**Competing interest.** The authors declare none.

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
