## [Reviewer Report]

The study presents a process-based model of (blueberry) fruit growth that integrates the quantitative effects of ABA with carbon and water import dynamics. The approach and the investigated fruit processes are interesting and contribute to an increased understanding of fruit growth and development.

As a general comment, I believe that interpretation of the scenario analyses should be a bit more constrained. The effects of the climatic conditions are limited to direct effects on fruit processes, whereas whole-plant effects (on e.g. plant water potential, sucrose content in phloem) are currently ignored. This should be more emphasized in the discussion. In addition, the objectives should be written more coherently, and the major objectives of the study made more explicit.

<b>Specific comments:</b>

*Introduction*

P3 L3-4: this first sentence has little value and can be removed

P3 L9: should you specify which type of fuit? e.g., “berry fruit” or “fleshy fruit”

P3 L14: “depending on genotypes” => “depending on species and genotype”

P4 L8: “These biocemical regulation drive” => “These biochemical regulations drive”

P5 L1-3 and L10-12: this is a conclusion rather than an objective

*Material and Methods*

P5 L22: how did you quantify “consistent sunlight exposure”?

P7 L11: ABA in heat units. Is there no effect of drought on ABA?

P8 L6 The quantitative response: it would be good to display this response in a graph.

P10 L10: What is R_f: fruit respiration?

P12 L2: “scaling factor” or “sealing factor”?

P12 L4: “the influence of ABA in” => “the influence of ABA on”

P14 L11: local or global sensitivity analysis?

P14 eq. 19: explain why there is a factor 10 in the equation.

P15 L8-10: did you take into account effects on stem water potential of this temperature increase?

P15 L11-14: did you include effects on stem water potential of these scenarios?

fig 1: water accumulation also affects osmotic pressure, but no arrow is drawn

*Results*

P15 L23,24, this sentence is grammatically incorrect. Additionally, add units for the MAE and RMSE values, and limit the number of decimals to 2.

Fig 2 and Fig S1: can you explain why the pattern of sugar uptake and respiration is oscillating (Fig S1A and B), whereas the resulting dry matter (Fig 2B) is smooth?

P16 L10: “Supplemental Figure S2C, D” => “Supplemental Figure S1C, D”.

Figure 3: add units in caption and or axes and preferably indicate the variables on the graph frames.

P17 L22-23: “were consistent across all scenarios”: this statement seems redundant as they were pre-defined constants in the fruit growth model.

Fig S2: what is the difference between frame D (water uptake) and F (water flux)?

P19 L12-16: is the effect on stem water potential included in the model?

P19 L24 - P20 L2: lower temperatures would also result in a less negative stem water potential, which would result in a comparably higher water uptake (see e.g. https://onlinelibrary.wiley.com/doi/full/10.1111/pce.12411,

or references in https://doi.org/10.1093/insilicoplants/diab038)

*Discussion*

P21 L8-9: are the ABA equations valid under a wide range of conditions? What if drought events occur?

P22 L8: check the word “minimum”, replace by lower (as you use “upper” as well?)

P23 L6: “the coefficient limited sensitivity” => “the coefficient had limited sensitivity”

P23 L15: “reflecting” => “reflected by”

P23 L17-21: several models also included the explicit calculation of stem water potential and phloem sugar concentration to link responses at the plant level to water fluxes to the fruit

*Conclusions*

P26 L13-15: adding whole-plant water relations, as these drive the fruit water and carbon import, would also increase usefulness of the model.

P26 L16-17: can you be more specific on how the model can help developing management approaches (e.g., which management decisions are taken and how can the model help).

---

## [Reviewer Report]

This manuscript develops a model of blueberry fruit development integrating hormonal, genetic and biophysical parameters. These are integrated using a series of equations and parameterized using empirically derived measurements.

The approach and methodology is very clearly explained, leaving little ambiguity as to what was done. The progressive introduction of additional variables, and the rationale for their inclusion, made this easy to follow.

The model shows good predictive power across a range of growing seasons demonstrating a relatively accurate representation of the system.

It would be beneficial to make it more explicit which parameters were derived from measurements and which were inferred, with more details on the latter process.

The integration of these results with other similar studies, contrasting the benefits of the unique elements of this model, would further improve the work. This could possibly be examined through the omission of key aspects of the model to demonstrate the need for their inclusion.

---

## [Editor Report]

Our sincere apologies for the delay in the revision of this manuscript.

There are two reviews which make a series of suggestions should you wish to revise the manuscript.

Best wishes

George

---

## [Reviewer Report]

There remains a lack of clarity as to which parameters were inferred and which were empirically derived. This clarity is important towards understanding how the model was generated.

While this might be the first model investigating ABA specifically, many models of fruit growth have been previously reported. Engaging more with this literature in the context of the work in this manuscript will strengthen the manuscript.

---

## [Reviewer Report]

Thanks for taking into account the comments and suggestions.

I found one typo in the legend of figure 5: “Acutal” should be “Actual”

---

## [Editor Report]

Dear Sun Woo Chung

Thanks for your revised manuscript. We are happy to consider this again with the minor edits suggested.

Regards

George